# Pathogenesis and Clinical Management of Uterine Serous Carcinoma

**DOI:** 10.3390/cancers12030686

**Published:** 2020-03-14

**Authors:** Li Zhang, Suet Ying Kwan, Kwong Kwok Wong, Pamela T. Soliman, Karen H. Lu, Samuel C. Mok

**Affiliations:** 1Department of Gynecologic Oncology and Reproductive Medicine, The University of Texas MD Anderson Cancer Center, Houston, TX 77030, USA; skwan1@mdanderson.org (S.Y.K.); kkwong@mdanderson.org (K.K.W.); psoliman@mdanderson.org (P.T.S.); khlu@mdanderson.org (K.H.L.); 2The University of Texas Graduate School of Biomedical Sciences at Houston, Houston, TX 77030, USA

**Keywords:** endometrial cancer, uterine serous carcinoma

## Abstract

Uterine serous carcinoma (USC) is an aggressive variant of endometrial cancer that has not been well characterized. It accounts for less than 10% of all endometrial cancers and 80% of endometrial cancer–related deaths. Currently, staging surgery together with chemotherapy or radiotherapy, especially vaginal cuff brachytherapy, is the main treatment strategy for USC. Whole-exome sequencing combined with preclinical and clinical studies are verifying a series of effective and clinically accessible inhibitors targeting frequently altered genes, such as *HER2* and *PI3K3CA*, in varying USC patient populations. Some progress has also been made in the immunotherapy field. The PD-1/PD-L1 pathway has been found to be activated in many USC patients, and clinical trials of PD-1 inhibitors in USC are underway. This review updates the progress of research regarding the molecular pathogenesis and putative clinical management of USC.

**Precis:** Uterine serous cancer, although rare, is the most lethal type of uterine cancer. It has distinct molecular features and pathogenic pathways compared with other uterine cancer types

## 1. Introduction

Endometrial cancer (EC) is the fourth most commonly diagnosed malignancy and the seventh most common cause of cancer death in women in the United States. Approximately 61,880 new cases were diagnosed and 12,160 EC-related deaths occurred in 2019, marking increases in incidence and mortality [1]. Worldwide, EC is the fourth most common tumor type and accounts for 5% of all cancer cases and 2% of all cancer deaths. EC’s highest incidence occurs in North America and Northern Europe, especially in the developed countries [1,2]. 

According to Bokhman’s 1983 model, EC is broadly classified based on histopathologic features into two categories, type I and type II, which differ in incidence, prognosis, epidemiology, molecular pathology, and clinical behavior (Table 1) [3]. Type I tumors, which account for 80% to 90% of all ECs, display a well-differentiated endometrioid histological phenotype and, compared with type II tumors, have a favorable prognosis, earlier onset, and higher 5-year survival rate [4]. Type I tumors are associated with obesity and usually develop from the environment of endometrial hyperplasia resulting from excess exposure to either endogenous or exogenous estrogen [4,5]. Type II tumors (nonendometrioid carcinomas), such as serous carcinoma, clear cell carcinoma, carcinosarcoma/malignant-mixed Müllerian tumor, and partial grade 3 endometrioid EC (EEC), are characterized by poorly differentiated histology and deep migration/invasion. However, in clinical practice, the classification of EC remains problematic due to inter-observer variability, particularly in high-grade carcinoma. Moreover, due to high heterogeneity, grade 3 EEC displays dissimilar features. Although a large number of grade 3 EEC cases show a serous-like phenotype, others display unequivocally endometrioid morphology [6,7]. Type II tumors are typically detected in women 70 years or older and carry a poor prognosis, high recurrence frequency, and much lower 5-year survival rate compared with type I tumors. In addition, type II tumors usually lack estrogen/progesterone receptors and do not respond to hormonal fluctuations [5]. 

Among the type II tumors, uterine serous carcinoma (USC) is the most common subtype. This highly aggressive variant accounts for only 10% of all ECs but 80% of all EC-associated deaths [8,9]. The 5-year tumor-specific survival rate is 74% and 33% for early- and late-stage USC patients, respectively (and 89% and 77% for low- and high-grade EEC) [10]. Unlike type I EC, the risk of metastasis and recurrence of USC does not rely on primary tumor size or grade; this trait contributes to a low overall survival rate of 18% to 27%. Complete surgical staging is performed, comprising total hysterectomy, bilateral salpingo-oophorectomy, and lymph node dissection, followed by carboplatin and paclitaxel. In light of poor patient survival and high recurrence rates, the development of targeted therapies specific to USC pathway aberrations would aid in its management. However, conducting studies that focus solely on this rare subtype has been a constant challenge, making it difficult to determine optimal treatment strategies [9]. 

**Table 1 cancers-12-00686-t001:** Characteristics of endometrial cancer.

Feature	Type I	Type II	References
Typical patient age, years	50–69	≥70	[4,8,11]
Hormone sensitivity	Yes	No	[4,8,11]
Precursor lesions	Atypical endometrial hyperplasia	Less defined	[4,8,11]
Subtypes	Endometrioid carcinoma and its variants	Uterine serous carcinoma and its variants	[4]
Behavior	Favorable/localized	Aggressive/prone to metastasis	[4]
Molecular alterations	MSI with MMR defects (20%)	*TP53* mutation (90%)	[1,8,11,12]
	*PTEN* deletion (80%)	HER2 overexpression (45%)	
	*CTNNB1* (40%)	*HER2* amplification (70%)	
	*PI3K* alteration (39%)		
Five-year survival	85%	43%	[4]

Abbreviations: MMR, DNA mismatch repair; MSI, microsatellite instability.

To better improve outcomes for women with EC, a new classification system based on The Cancer Genome Atlas (TCGA) EC dataset was recently developed. This classification system includes 4 distinct EC subgroups with distinct genomic aberrations [13]. The first subgroup is DNA polymerase epsilon (*POLE)*-mutant EC, which consists of copy number-stable, but ultra-mutated, ECs with recurrent mutations in the exonuclease domain of *POLE*. The second subgroup is microsatellite instability (MSI)-high EC, which consists of hyper-mutated ECs with MSI due to dysfunctional DNA mismatch repair (MMR) proteins MLH1, MSH2, MSH6, and PMS2. The third subgroup consists ECs that are MMR proficient and have mutations in genes associated with the PI3K/Akt and Wnt signaling pathways. The fourth subgroup consists of ECs that are similar to high-grade serous ovarian cancer and have high frequencies of somatic copy number alterations and *TP53* mutations. This subgroup includes both grade 3 EECs and USCs.

## 2. Histopathology of USC

Most type II ECs, especially USC, have a complex papillary or glandular architecture, which is similar to serous papillary ovarian carcinomas. In general, densely fibrotic papillae fronds and slit-like spaces are common. Huge, round, undifferentiated tumor cells with a high number of mitotic figures, increased nuclear-to-cytoplasmic ratios, and prominent nuclear atypia are detached and present in the blank spaces. Hobnail cells, clear cells, and polygonal cells are frequently observed. Classic psammoma bodies and cilia are also observed in some cases, as in papillary serous ovarian carcinomas [14,15]. 

USC is commonly considered to be derived from its putative precursor, endometrial intraepithelial cancer, a lesion characterized by the malignant transformation of either endometrial surface epithelium or underlying glands [16]. However, this concept has been challenged by many studies demonstrating that endometrial intraepithelial cancer is also associated with extra-uterine serous carcinoma; furthermore, endometrial glandular dysplasia has been proposed as a USC precursor [17,18]. 

## 3. Molecular Pathogenesis of USC

Thanks to various high-throughput techniques such as whole-exome sequencing, DNA microarray, mRNA fingerprints, hierarchical cluster analysis, and proteomic characterization, the molecular profiles associated with clinical-pathological performance of type I and type II tumors have been revealed [19,20,21]. In type I EC, unopposed estrogen is considered the predominant driver for tumor initiation, and deregulated balance of pro-growth estrogens and anti-growth progestogens is the potent trigger of the disease [22]. Type II EC is barely affected by the estrogen pathway due to the lack of estrogen receptor (ER) expression [12]. The genetic factors that contribute to type I and type II EC are quite different [12]. Alterations of MMR genes *MLH1* and *MSH6* exist in about 33% of type I EC patients; these alterations drive non-atypical hyperplasia to complex atypical hyperplasia [23,24]. *PTEN* mutation occurs in approximately 40% to 60% of type I ECs, and both *MLH1* and *MSH6* alterations are considered to be almost exclusively restricted to type I EC [25].

In type II EC, the predominant alteration is *TP53*, which accounts for up to 95% of cases and plays essential roles in almost all the disease’s stages, especially the early stage [26]. *TP53* alteration occurs in less than 10% of type I ECs, suggesting distinctive effects of *TP53* in these two types of EC [25,26] ( Figure 1). Among type II ECs, USC has been found to harbor a 60% *TP53* mutation rate (Figure 2). Although p53 immunohistochemistry, which is easy and inexpensive for pathologists to perform, has evolved into a commonly used tool to aid the diagnosis of various cancers, due to the problems in interpretation of staining results, the feasibility of using p53 staining in EC diagnosis needs to be further evaluated [27]. Based on literature, diffuse strong nuclear accumulation involving more than 80% of EC tumor cells is the typical staining pattern for *TP53* missense mutations and is more likely to be observed in USC than in low-grade EC subtypes [28]. However, the presence of p53 nuclear accumulation is not always linked to *TP53* gene mutation [28]. 

Schematic descriptions of the initiation and development of high-grade endometrioid (blue) and uterine serous carcinoma (yellow) from benign endometrium and factors that potentially drive these processes.

In addition to *TP53*, mutations in other tumorigenesis-relevant genes, including *PPP2R1A, PIK3CA*, *PIK3R*, *HER2, FBXW7*, *CHD4,* and others have been identified in USC patients [15,29,30]. The mutation frequencies of the top 20 genes in USC are summarized in Figure 2, and the mutational gene profiles of 206 USC patients are summarized in Figure 3. 

### 3.1. Metabolic Profile Alterations 

Alterations of cancer cell metabolic features are frequently observed in various cancer types, making metabolic reprogramming one of the key hallmarks of tumorigenesis [32]. With the explosion of cancer metabolism studies in recent decades, clinically accessible inhibitors of energy metabolism in cancer cells are increasingly emerging, and some have already received promising data from clinical trials [33]. There are two main categories: (1) small molecules against fundamental enzymes in different metabolic pathways, such as glucose transporters (GLUTs) in glucose uptake, hexokinase 2 in glycolysis, and citrate dehydrogenase in the TCA cycle [34,35,36] and (2) competitive molecules of essential metabolites such as 2-deoxy-D-glucose [37]. Studies focused on metabolic alterations are rare in EC, let alone USC. The first study to reveal disturbed energy metabolism in EC can be traced back to Benjamin and Romney in 1964 [38]. Since then, other metabolic features have been identified in EC. 

#### 3.1.1. Glycolysis 

Byrne et al. examined metabolic vulnerabilities in EC [39]. They found that the glycolysis-lipogenesis process was dramatically elevated in their pool of human EC cell lines compared with non-tumorous endometrial tissue-derived immortalized cell lines. Altered protein expression involved in glycolysis-lipogenesis was identified by a microarray using eight endometrial tissue samples (four with and four without type I EC); in these, GLUT6 showed a highly increased expression pattern [39]. GLUTs are the entrance for glycose consumption and usually exhibits an elevated expression in malignant cells [40]. 

In addition, GLUT1 is highly increased in EC [41]. Pathologists have explored GLUT1 expression patterns in type I and II EC immunohistochemically. GLUT1 protein levels are much higher in tumor tissues than in normal tissue and correlate with clinic-pathologic variables in different patient cohorts, implying that GLUT1 is both a prognostic and diagnostic marker for EC patients [42]. Moreover, upregulation of GLUT1 was found to coincide with increasing grade of EC. Another GLUTs family member, GLUT8, had increased expression in all EC subtypes compared with atrophic endometrium. GLUT1 and GLUT8 were shown to display a stepwise increased expression as EC histopathology worsens [43]. However, no studies of GLUTs in USCs have been reported to date. 

#### 3.1.2. Mitochondrial Function

Using RNA sequencing profiles of 271 EEC patients obtained from the TCGA database, Liu et al. performed consensus unsupervised clustering analysis on 2786 genes [44]. Four EEC clusters with different transcriptome profiles were identified, among which cluster II (*n* = 61) contained young patients with low-grade and early-stage EEC, high risk of recurrence, and poor survival outcomes. Differential genes were identified in cluster II compared with other clusters. Ingenuity pathway analysis (IPA) was used to identify metabolic pathway enrichment. Gene ontology enrichment and gene set enrichment analysis were also performed. After an in-depth statistical analysis using the Mann–Whitney test, suppression of the TCA cycle was found in cluster II and positively correlated with *PD-L1* gene expression, suggesting a role of mitochondrial function in immune resistance [44]. However, no similar study has been performed in USC, most likely due to the limited number of USC cases.

The new application of metformin in EC treatment, especially in USC, reveals that the electron transport chain complex I (NADH: ubiquinone oxidoreductase), located in the mitochondrial membrane, is a viable pharmacological target [45]. Currently, it is widely accepted that metformin specifically targets complex I, which catalyzes the first step of the mitochondrial electron transport chain and is mainly responsible for oxidizing NADH to NAD+ and establishing the hydrogen ion gradient [45]. Multiple in vitro studies with USC cell lines demonstrated that metformin inhibited cell proliferation and metastasis via inhibiting oxidative phosphorylation (OXPHOS) and ATP consumption, further activating AMPK to suppress its downstream targets such as the mTOR and STAT3 pathways [46]. Other important signaling pathways were also identified, such as PI3K/AKT/mTOR and MAPK/ERK [46,47]. Several clinical trials and case reports of metformin treatment as a single agent or combination with other treatments have been established; for example, a phase II/III trial will add metformin or placebo to paclitaxel/carboplatin as the first-line therapy for advanced EC (NCT02065687) [45,48]. Some studies have produced promising data. For example, a multi-institutional retrospective cohort analysis of 1495 EC patients demonstrated that metformin exposure improved recurrence-free and overall survival [49]. 

Diverse mitochondrial DNA mutations and increased mitochondrial biogenesis were well identified in type I EC by a series of publications [50]. Mutations in complex I-associated genes have been observed in 70% of type I ECs [50]. However, no such study has focused on USC to date. 

### 3.2. Epigenetic Alterations

Epigenetic abnormalities of key factors associated with carcinogenesis are also commonly observed in USC. These include EZH2, DNA methylation, and noncoding RNAs. Reversal of epigenetic alteration is considered to be a promising strategy for cancer treatment [51]. Extensive efforts have been made to identify the promoter methylation profile in USC. However, it has been demonstrated that promotor methylation plays a much smaller role in USC progression than in EEC [52]. The methylation level is much lower in USC than in EEC partially due to reduced expression of DNMT1 and DNMT3B [52,53]. In addition, although several noncoding RNAs (ncRNAs) have been implicated in EC progression, only a few regulatory ncRNAs have been shown to play a role in USC [54]. For instance, NEAT1 contributes to the aggressiveness and progression of ECs, including USC, by serving as an oncogenic sponge of microRNA-361 [55], and MEG3 has been shown to modulate USC cell proliferation by directly inhibiting PI3K [56].

### 3.3. Signaling Pathway Crosstalk in USC

The identification of genetic and epigenetic alterations allowed the discovery of crosstalk between key signaling pathways in USC cells, which confers the malignant phenotype of USC (Figure 4). Interactions between the two key signaling pathways, namely the PI3K/AKT/mTOR and p53 signaling pathways, can be linked to the energy-related AMPK pathway and enzymes involved in glycolysis and OXPHOS, which subsequently lead to enhanced ETC activity and increased ATP production.

## 4. Clinical Management of USC

### 4.1. Diagnosis

Similar to the other subtypes of EC, the earliest symptom of USC typically is postmenopausal vaginal bleeding [57]. Patients with potential USC undergo pelvic examination for abnormalities in the uterus, vagina, ovaries, fallopian tubes, bladder, and rectum [15]. A series of microscopy-based tests such as hysteroscopy, cystoscopy, proctoscopy, and dilation and curettage of the uterus are also commonly used. Blood tests such as the CA-125 test are routinely used for USC diagnosis. Examining the endometrial biopsy specimen is a sensitive and efficient diagnostic approach for USC, although in some cases USC is mixed with other histologies [58]. The Pap SEEK test can screen for gene mutations and chromosome alterations that frequently occur in USCs, such as *TP53, FBXW7, PIK3CA,* and *PIK3R* mutations, by using a trace of DNA from uterine tissue acquired in a Papanicolaou test [59]. 

### 4.2. Treatment Approaches

#### 4.2.1. Surgical Staging 

Currently, the most common approach for USC treatment is surgery followed by chemotherapy and radiotherapy [60]. Similar to ovarian cancer, the routine and comprehensive surgical treatment and staging of USC includes total hysterectomy, bilateral salpingo-oophorectomy, bilateral pelvic lymphadenectomy, systematic para-aortic lymphadenectomy, complete omentectomy, and peritoneal cytology [61]. Minimally invasive surgeries are also regularly performed [62,63]. Due to the high heterogeneity and high metastatic potential of USCs, surgical management needs to be optimized according to the stage, histological subtype, tumor size, tumor location, severity of symptoms, metastasis status, and patient’s health status [62,64]. 

#### 4.2.2. Chemotherapy and Radiotherapy

After surgery, adjuvant chemotherapy is routinely recommended to USC patients, with the purpose of killing the remaining cancer cells or preventing them from growing [65]. However, for cases in which USC arises from a polyp without myoinvasion, adjuvant therapy or postoperative observation is equally recommended on the basis of studies by Mandato et al. and Thomas et al. [66,67]. These studies suggest that adjuvant therapy can be avoided if no residual tumor is left in the uterus after surgical staging [66,67].

Platinum/taxane-based adjuvant chemotherapy is the most commonly used regimen in both early- and late-stage USC patients [68]. Clinical trials that treated USC with drug combinations such as carboplatin-paclitaxel and doxorubicin-cisplatin-paclitaxel achieved promising results: these combinations significantly extended survival and decreased recurrence rates (NCT00231868, NCT00147680, NCT00052312, and NCT00052312) [57,69]. 

In most cases, radiotherapy is used together with chemotherapy to achieve a better outcome [70]. Two kinds of radiotherapy have been reported in USC: whole-abdomen radiotherapy with a pelvic boost and vaginal cuff brachytherapy [71]. 

#### 4.2.3. Targeted Therapies

The high rate of side effects (e.g., white blood cell toxicity, alopecia, fibrosis in the intestines, hematuria, cystitis, bone marrow effects) of conventional therapies for USC should be noted [72]. Furthermore, recurrent USC is less responsive to chemotherapy compared with other EC subtypes. The good news is that those problems are likely to be solved by molecular targeted drugs, and therapeutic agents targeting the PI3K/AKT/mTOR signaling pathway, cell cycle regulation, and the programmed cell death protein 1/programmed death ligand 1 (PD-1/PD-L1) pathway have already been exploited in USC (Figure 5) [73]. However, the study of targeted therapy for USC lags behind that of other cancer types. Hormonal treatment, which is the only targeted therapy approved by the US Food and Drug Administration (FDA) for EC, is only applied for hormone-dependent EEC [74]. Multiple clinical trials targeting advanced-stage ECs, including USC, have been launched (Table 2). However, due to the rarity of USC, only two trials (NCT01367002 and NCT03285802) were designed specifically for USC patient cohorts.

##### p53 Signaling Pathway Inhibitors

As the most altered molecule in USC, p53 and the p53-associated signaling pathway are promising clinical targets. Currently, therapies targeting the p53 pathway focus on either restoring wild-type p53, “correcting” mutated p53 activities, or targeting the downstream effector of p53 [75]. For instance, PRMA-1, a small molecule, has been demonstrated to restore wild-type p53 activity through changing mutated p53 conformation [76]. Furthermore, p53-mutated USC cells are sensitive to combination treatment with EGFR inhibitors gefitinib and paclitaxel and treatment with polo-like kinase 1 inhibitor BI2536 [77]. Moreover, the combination of proteasome and histone deacetylase inhibitors can overcome the effect of p53 mutations [78].

##### HER2/Neu Inhibitors

Although trastuzumab (anti-HER2/neu antibody) has been examined in USC, clinical trials showed no significant effects in USC patients after treatment with trastuzumab alone [79]. Combination therapies such as trastuzumab with chemotherapy showed much better clinical outcomes in a series of case reports and clinical trials. In a randomized phase II trial of carboplatin-paclitaxel vs carboplatin-paclitaxel-trastuzumab in 61 USC patients, chemotherapy plus trastuzumab was well tolerated without unexpected toxicity and achieved a significant increase in progression-free survival [69]. Ado-trastuzumab emtansine (T-DM1) is a pharmacological conjugate that contains trastuzumab and an anticancer drug named DM1. T-DM1 demonstrated encouraging antitumor activity in USC cell lines and USC xenografts [80]. A phase II clinical trial evaluating the effect of T-DM1 in solid tumors is ongoing (NCT02675829). An optimized antibody-drug conjugate, SYD985, was also evaluated in USC and shows higher efficacy than T-DM1, especially in cells with low or moderate HER2/neu expression [81].

Tyrosine kinase inhibitors are proven to have efficacy in USC. The most commonly used is lapatinib, a reversible dual inhibitor of HER2 and EGFR [82]. Although some in vitro data showed the suppressive effect of this drug in USC cell lines, several clinical trials have demonstrated limited clinical activity of single-agent lapatinib treatment [83,84]. Combination treatment with lapatinib and trastuzumab achieved strong synergistic antitumor activity in HER2/neu-overexpressing USC xenograft tumors and USC patient-derived xenografts [83]. Furthermore, lapatinib showed promising results in a cohort of EC patients with E690K mutation in *EGFR* [84].

Anti-HER2 vaccines have been well established in clinical investigations of various HER2/neu-expressing tumors [85,86]. However, studies of anti-HER2 vaccines in USC are lacking. 

##### PI3K/AKT/mTOR Signaling Pathway Inhibitors 

According to the updated literature, there are four categories of inhibitors targeting PI3K/AKT/mTOR signaling pathway: mTOR inhibitors, PI3K inhibitors, dual mTOR/PI3K inhibitors, and AKT inhibitors. Among these, drugs that block mTOR activity are the most studied [87]. 

(1) mTOR Inhibitors

Rapamycin and its analogues (rapalogs, e.g., sirolimus, temsirolimus, everolimus), which inhibit mTORC1 activity, are considered the first clinical PI3K signaling inhibitors [88]. Although temsirolimus is FDA approved for treating advanced renal cell carcinoma and everolimus is approved for the treatment of advanced breast cancer, nonfunctional gastrointestinal and lung neuroendocrine tumors, and renal cell carcinoma, rapalogs are not yet approved for the treatment of EC and are under development [89,90,91]. 

Currently, several phase II clinical trials of signal-agent rapalog treatment in mixed cohorts of EC patients, including those with EEC, USC, and clear cell endometrial cancer, have been completed (NCT00087685, NCT00072176, and NCT00122343). Taken together, these trials show that single-agent rapalog treatment has modest, acceptable, and reproducible antitumor activity across EC subtypes. Further trials are underway to determine optimal combined treatments. For instance, in an ongoing phase I trial, γ-secretase/Notch signaling pathway inhibitor RO4929097 was tested together with temsirolimus in 18 patients with advanced solid tumors, including USC (NCT01198184). Another open-label treatment program for patients with recurrent USC combines everolimus and the hormonal drug letrozole (NCT03285802). 

To overcome the limitations of rapalogs, the so-called second generation of mTOR inhibitors that dually inhibit kinase activities of mTORC1 and mTORC2—which include AZD8055, OSI027, and INK128 (MLN0128)—have been studied extensively [92]. An in vitro study by English et al. illustrated that AZD8055 strongly suppressed the proliferation of 22 primary USC cancer cell lines by arresting cells in the G0/G1 phase [93]. Of note, a single-agent phase I clinical trial of INK128 in advanced malignancies, including USC, was initiated in 2009 (NCT01058707). Another phase I clinical trial led by Dana-Farber Cancer Institute used the combined intervention of MLN0128 and bevacizumab (an anti-angiogenesis drug) to treat patients with solid carcinomas, including endometrial clear cell adenocarcinoma and USC (NCT02142803). Other combination strategies in EC, such as MLN0128 plus MLN1117 (a PI3K inhibitor) and MLN0128 plus paclitaxel, are under investigation in a phase II trial (NCT02725268). 

(2) PI3K Inhibitors

Another key therapeutic target is PI3K. Drugs targeting either a single isoform of PI3K (isoform-specific inhibitors) or all four isoforms (pan-PI3K inhibitors) are available [94]. Pan-PI3K inhibitors were the first generation of PI3K inhibitors, comprising GDC-0941, BKM120, PX866, ZSTK474, and BAY80-6946 (copanlisib) [94]. The first phase I clinical trial of BKM120 in patients with solid tumors, including EC, demonstrated an encouraging clinical outcomes: preliminary antitumor activity, acceptable toxicity, and favorable pharmacokinetic profile (NCT01068483) [95]. BKM120 treatment in advanced or recurrent EC has entered into a phase II clinical trial (NCT01289041). Copanlisib also showed favorable clinical outcomes in a phase I trial for patients with advanced solid tumors [96]. After that, phase II trials of copanlisib in patients with EC opened (NCT02728258, NCT03586661). 

Isoform-specific PI3K inhibitors—especially those that target the PI3Kα subunit, which harbors the majority of PI3K mutations in solid tumors—are another promising therapeutic option for USC [73]. For instance, preclinical and clinical studies of taselisib (GDC-002), a *PIK3CA* mutation-selective inhibitor, indicated its antitumor functions in ECs (NCT01296555) [97,98]. 

(3) mTOR and PI3K Dual Inhibitors

mTOR and PI3K dual inhibitors are also applied for USC treatment and have the benefit of blocking the whole signaling pathway without inducing complicated feedback loops, which are often observed in certain malignancies treated with a single inhibitor [99]. The design of dual mTOR and PI3K inhibitors is based on the high-sequence homology of the catalytic region sites in both mTOR and PI3K [87]. The first-in-human phase I clinical trial of the mTOR and PI3K dual inhibitor LY3023414 in patients with advanced solid tumors, including EC (*n* = 15), was completed by Bendell et al. in 2017, and the drug demonstrated strong antitumor activity with favorable safety and pharmacokinetic profiles [100]. In patients with *PI3KR1-* and *PTEN*-mutated EC, a durable response to LY3023414 was observed [100]. A phase II clinical trial of LY3023414 for treatment of recurrent or persistent EC is ongoing (NCT02549989). Other mTOR and PI3K dual inhibitors, such as apitolisib (GDC-0980) and NVP-BEZ235, are also under clinical investigation (NCT01455493, NCT01195376). 

(4) AKT Inhibitors 

AKT mutation is rarely observed in USC, and fewer AKT inhibitors have been studied in these tumors compared with PI3K and mTOR inhibitors. Currently, AKT inhibitors either compete for the ATP binding subunit—as do isoquinoline-5-sulfonamides, azepane derivatives, and thiophenecarboxamide derivatives—or inhibit AKT allosterically—as do 2,3-diphenylquinoxaline, alkyl-phospholipids, and purine derivatives [101]. Some studies of AKT inhibitors have included USC [102]. For instance, preclinical data for MK-2206, an allosteric inhibitor of AKT, demonstrated remarkable suppressive effects in three distinct patient-derived xenografts: USC1 (uterine serous), EEC2 (endometrioid grade 2), and EEC4 (endometrioid grade 3) [103]. Clinical trials of MK-2206 in ECs, including USC, provided valuable evidence for future clinical applications (NCT01312753, NCT01307631) [104]. 

In addition to single-agent treatment or combined therapies with drugs in the same category, considerable efforts have been made to design new strategies of blocking the PI3K/AKT/mTOR pathway to improve therapeutic efficacy and prevent adverse effects and stubborn resistance [105]. For example, a phase I clinical trial studied the clinical outcomes of dually targeting the PI3K/AKT/mTOR and RAF/MEK/ERK pathways in 236 patients with advanced solid tumors, including EC [106]. Dual inhibition exhibited more favorable efficacy compared with single inhibition and may be important for USC patients with mutations in the PI3K signaling pathway and KRAS or BRAF [106]. Dual inhibition of HER2/PIK3CA was also shown to overcome single treatment-related drug resistance significantly in HER2-amplified USC cells and xenografts [107]. 

#### 4.2.4. Cyclin-Dependent Kinase Inhibitors

A high percentage of alterations in cell cycle-related genes has been observed in USC, and mutations affecting the Fbxw7/Cyclin E pathway are the most frequent in USC [13]. Four generations of cyclin-dependent kinase (CDK) inhibitors have been synthetized, and some are under clinical investigation in patients with EC. For example, a phase I study of ribociclib (LEE011), a selective inhibitor of CDK4/6, demonstrated an acceptable safety profile and preliminary antitumor activity in patients with advanced tumors including EC [108]. A trial of combined treatment with ribociclib, everolimus, and letrozole in patients with advanced or recurrent EC is still recruiting (NCT03008408). CYC065, an inhibitor of CDK2/9, alone or together with taselisib, shrank USC xenografts, which were derived from a USC harboring *CCNE1* amplification and *PIK3CA* mutation [109]. However, to date, no clinical trial has evaluated the efficacy of CDK inhibitors specifically in patients with USC. 

### 4.3. Immune Profiling and Immunotherapy 

Studies of immunotherapy in EC focus on inhibiting immune checkpoints, which is currently the most promising approach in immunotherapy [110]. The most studied immune checkpoint in EC is the PD-1/PD-L1/2 pathway [111]. PD-1 (also known as CD274), a member of B7-CD28 family, is a 55 KD transmembrane protein and is induced in activated immune cells such as CD4+, CD8+, and CD4-CD8-T cells as well as activated dendritic cells and macrophages [112]. PD-1 has 2 ligands, PD-L1 (CD279; B7-H1) and PD-L2 (CD273; B7-DC). PD-L1 is the predominate one and is expressed on the surface of a large spectrum of immune cells and non-hematopoietic cells, such as vascular endothelial cells, to protect those tissues [112]. 

Many studies have illustrated a high expression pattern of PD-L1/2 in various EC subtypes [113]. The high expression of PD-L1/2 has been shown to be correlated with poorly differential status of EC subtypes, poor prognosis, poor diagnosis, and low survival rate [114]. Mo et al. evaluated PD-1, PD-L1, and PD-L2 expression by immunohistochemistry in 35 human normal endometrium samples and 75 human ECs (63 EEC, 11 USC, one clear cell). The results showed that the expression of PD-L1 in tumor-infiltrating immune cells was associated with the histology of EC: the percentage of PD-L1-positive cells in the tumor-infiltrating immune cells was much higher in non-endometrioid ECs than endometrioid ones [114]. 

Pembrolizumab, an FDA-approved PD-1 inhibitor, has been applied to treat EC in several preclinical studies and clinical trials [115]. In the phase I KEYNOTE-028 clinical trial, 23 EC patients were enrolled, including 17 with EECs, three with other adenocarcinomas, two with USCs, and one with carcinosarcoma (NCT02054806). A 13% overall response rate was achieved in this patient population [116]. Due to the small number of USC patients recruited in this study, the response rate for immune checkpoint inhibitor treatment in these patients remains unclear.

**Table 2 cancers-12-00686-t002:** Summary of clinical trials of targeted therapies in uterine serous carcinoma.

Drug	Function	Treatment Regimen	Phase	Status	Patient Cohort (EC Including USC)	ClinicalTrials.Gov Identifier	References
Trastuzumab	Anti-HER2/neu antibody	IV over 30–90 min on days 1, 8, 15, 22. Courses repeat every 28 days.	II	Completed	Stage III, IV, or recurrent EC with HER2/neu amplification (*n* = 34)	NCT00006089	N/A
Trastuzumab-IL-12	Trastuzumab: anti-HER2/neu antibody IL-I2: cytotoxic lymphocyte maturation factor	Trastuzumab: IV on day1, with maintenance dose on day 1 of each subsequent week. IL-I2: IV on days 2, 5 from week 3.	I	Completed	Recurrent cancers with high HER2/neu (*n* = 100)	NCT00004047	[117]
Trastuzumab-paciliatxel-IL-I2	Trastuzumab: anti-HER2/neu antibody IL-I2: cytotoxic lymphocyte maturation factor	Course 1: Trastuzumab IV on days 1, 8, 15; paclitaxel IV on day 1. Course 2: course 1 plus IL-12 SQ on days 2, 5, 9, 12, 16,19. Courses to be repeated q21 days.	I	Completed	Recurrent solid tumors (*n* = 18)	NCT00028535	N/A
Trastuzumab-carboplatin-paclitaxel	Anti-HER2/neu antibody	Paclitaxel: 175 mg/m^2^ for 21 days for 6 cycles. Carboplatin: AUC 5 for 21 days for 6 cycles. Trastuzumab: 6 mg/kg for 21 days for 6 cycles from day 21.	II	Active, not recruiting	Stage III-IV or recurrent USC with HER2/neu amplification (*n* = 61)	NCT01367002	[69]
Lapatinib	Dual tyrosine kinase inhibitor of HER2/neu and EGFR	PO once daily on days 1–28. Courses repeat every 28 days.	II	Completed; Has Result	Recurrent EC (*n* = 31)	NCT00096447	[118]
Lapatinib-ixabepilone	Lapatinib: inhibitor of HER2/neu and EGFR; ixabepilone: antimicrotubule agent	Lapatinib: 500–1250 mg PO once daily. Cycle every 21 days for 6 cycles+ ixabepilone 32 mg/m^2^ every week.	I	Unknown	Recurrent EC with high HER2/neu	NCT01454479	[26]
RAD001	mTOR inhibitor	10 mg PO daily.	II	Completed; Has Result	Progressive or recurrent EC (*n* = 35)	NCT00087685	[119]
Temsirolimus	mTOR inhibitor	Temsirolimus: IV over 30 min on days 1, 8, 15, 22. Courses repeat every 28 days.	II	Completed; Has Result	Metastatic or locally advanced recurrent EC (*n* = 62)	NCT00072176	[120]
Temsirolimus-RO4929097	Temsirolimus: mTOR inhibitor; RO4929097: γ-secretase/Notch signaling pathway inhibitor	Temsirolimus: IV over 30 min on day1- 6 (course 1 only). Temsirolimus IV or PO on days 1, 8, 15 and RO4929097 PO once daily on days 1–3, 8–10, and 15–17. Courses repeat every 21 days.	I	Completed	Advanced solid tumors (*n* = 18)	NCT01198184	[121]
Everolimus-letrozole	Everolimus: derivative of rapamycin, mTOR inhibitor; letrozole: aromatase inhibitor	Letrozole: 2.5 mg daily every 30 days. Everolimus: 10 mg daily every 28 days.	II/III	Active, not recruiting	Recurrent USC with *PIK3CA* gene mutation (*n* = 1)	NCT03285802	N/A
MLN0128-bevacizumab	TORC1/2 inhibitor	INK128: PO daily on days 1–28 and bevacizumab IV on days 1 and 15. Courses repeat every 28 days.	I	Active, not recruiting	Recurrent glioblastoma and other solid tumors (*n* = 58)	NCT02142803	[122]
MLN0128-MLN1117-paclitaxel	MLN0128: dual TORC1/2 inhibitor; MLN1117: PI3Kα inhibitor	Paclitaxel: 80 mg/m^2^ IV, weekly on days 1, 8, and 15 of a 28-day cycle. Paclitaxel 80 mg/m^2^ IV, weekly on days 1, 8, and 15 of a 28-day cycle along with MLN0128 4 mg capsule PO on days 2–4, 9–11, 16–18, and 23–25 of a 28-day cycle. MLN0128 30 mg capsule PO once weekly on days 1, 8, 15, and 22 of a 28-day cycle. MLN0128 4 mg capsule PO MLN1117 200 mg capsule PO on days 1–3, 8–10, 15–17, and 22–24 of a 28-day cycle.	II	Active, not recruiting	Advanced, recurrent or persistent EC (*n* = 245)	NCT02725268	[94,123]
BKM120	Pan-PI3K inhibitor	100 mg/day PO as a second-line therapy.	II	Completed	Advanced EC (*n* = 70)	NCT01289041	[124]
LY3023414	mTOR and PI3K dual inhibitor	RP2D of 200 mg PO twice daily.	II	Active, not recruiting	Recurrent or persistent EC (*n* = 31)	NCT02549989	N/A
GDC-0980	mTOR and PI3K dual inhibitor	PO daily.	II	Completed	Recurrent or persistent EC (*n* = 56)	NCT01455493	[125]
NVP-BEZ235	mTOR and PI3K dual inhibitor	BEZ235: dose escalation PO once daily.	I	Completed	Adult Japanese patients with advanced solid tumors (*n* = 35)	NCT01195376	[126]
MK2206	AKT inhibitor		II	Completed	Recurrent or persistent EC (*n* = 37)	NCT01312753	[127]
Ribociclib (LEE011)-everolimus-letrozole	Ribociclib: CDK4/6 inhibitor; everolimus: mTOR inhibitor; letrozole: aromatase inhibitor	Ribociclib: 250 mg PO daily for a 28 day cycle. Everolimus: 2.5 mg PO daily for a 28-day cycle. Letrozole: 2.5 mg PO daily for a 28-day cycle.	II	Recruiting	Malignant neoplasms of female genital organs; endometrial carcinoma (*n* = 76)	NCT03008408	[128]

Abbreviations: AUC, area under the curve; BID, twice a day; EC, endometrial cancer; IV, intravenously; PO, orally; RP2D, recommended phase II dose; USC, uterine serous carcinoma.

## 5. Conclusions

Studies of USC were lacking for a long time due to its rare occurrence, its prevalence in women over the age of 70 years, and the lack of animal models for the disease. Therefore, the development of treatments for this cancer is far behind those for other cancers. However, USC’s aggressive features, resistance to chemotherapy, poor prognosis, and extremely high recurrence rate (50% to 80%) contribute to the increase in EC-related deaths every year. Currently, surgery together with chemotherapy and radiotherapy remains the dominant treatment option for USC. 

Clearly understanding the molecular alterations in different EC subtypes facilitates the generation of accurate diagnosis and prognosis methods, as well as targeted therapeutic strategies. Current studies of somatic genomic profiles in USC are mostly performed together with other EC subtypes, and the number of USC patients is relatively small. In addition to treatments targeting single somatic hotspot gene mutations, emerging advances in tumor metabolism and immunotherapy have opened new doors for USC patients. In addition, systematic biochemical studies will help to synthesize novel therapeutic inhibitors. A much larger number of USC cohorts with varying patient populations will need to be recruited for clinical trials to evaluate the efficacy of new treatments. 

## Figures and Tables

**Figure 1 cancers-12-00686-f001:**
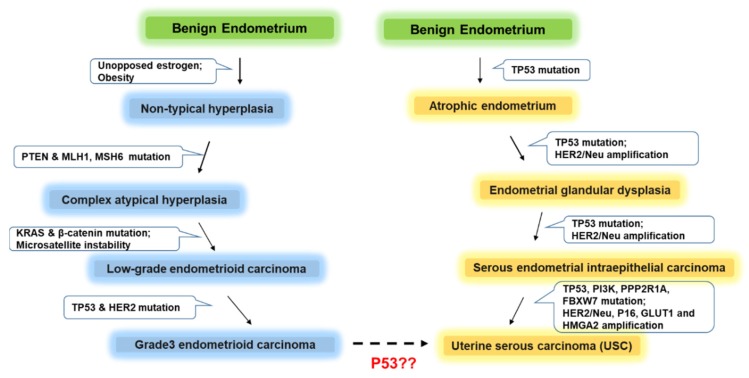
Molecular pathogenesis of endometrioid and uterine serous carcinoma.

**Figure 2 cancers-12-00686-f002:**
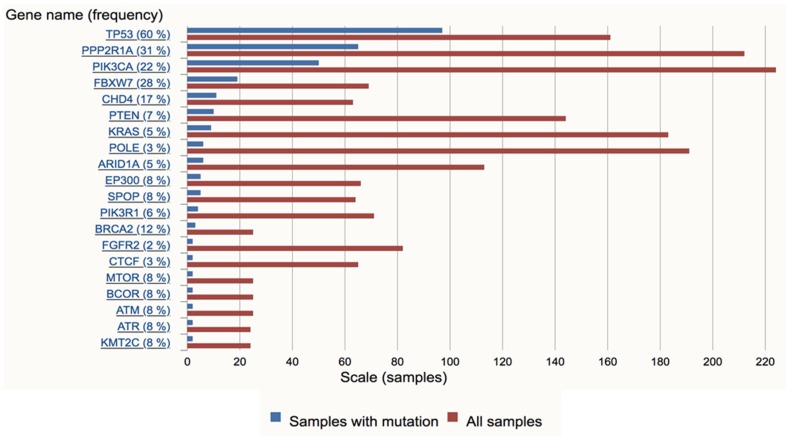
The 20 most frequently mutated genes in uterine serous carcinoma. Data were extracted from the COSMIC v90 database [31].

**Figure 3 cancers-12-00686-f003:**
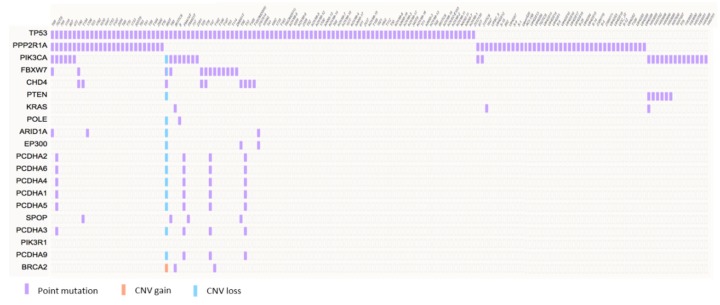
Mutation matrix of 206 uterine serous carcinomas. Data were extracted from the COSMIC v90 database [31], and the 20 most frequently mutated genes in uterine serous carcinoma are presented. Each column represents a single sample.

**Figure 4 cancers-12-00686-f004:**
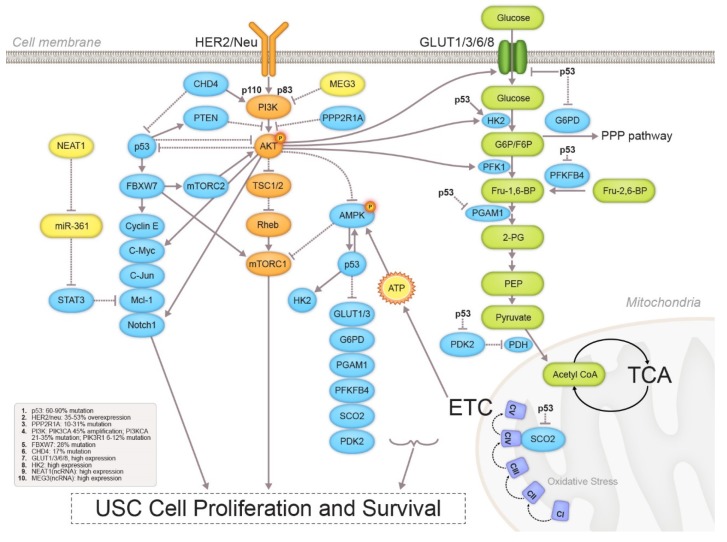
Summary of altered molecular pathways and their communications in USC.

**Figure 5 cancers-12-00686-f005:**
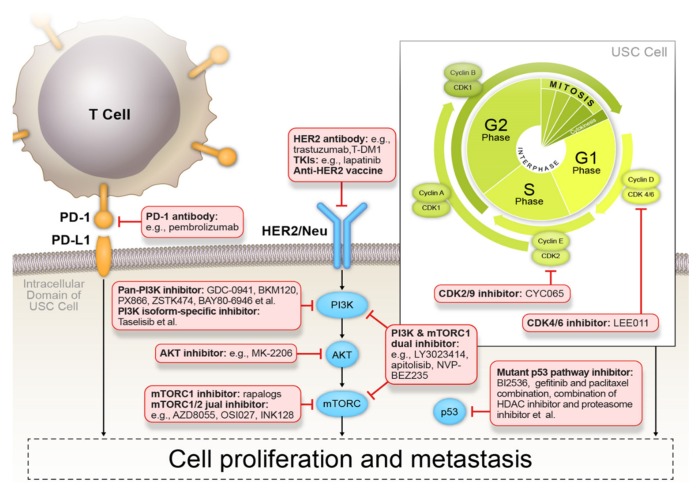
Molecular targeted therapy in uterine serous carcinoma. Reported drugs targeting the PI3K/AKT/mTOR signaling pathway, cell cycle regulation, and the PD-1/PD-L1 pathway are under clinical investigation for the treatment of patients with uterine serous carcinoma.

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
