# Peer review of "Pathogenesis and Clinical Management of Uterine Serous Carcinoma"

_cancers, 2020, doi:10.3390/cancers12030686_

Round 1

Reviewer 1 Report

Major issues:

Epigenetic changes (such as EZH2, DNA methylation, microRNAs and long non-coding RNAs) found in serous endometrial cancers also play critical roles in the tumorigenesis and metastasis are not fully mentioned. Necessary references should be added and discussed (For example, Cancers (Basel). 2019 Feb 16;11(2). pii: E234) . Either restoring wild-type p53 activity or inhibiting mutant p53 oncogenic activity may be an efficient strategy to treat serous endometrial cancers. In this review, the authors should summarize recent advances in the development and clinical trials of small molecules and compounds that aim to subvert oncogenic activities of mutant p53 protein into wild-type p53 tumor suppressor functions (For example, J Exp Clin Cancer Res. 2018 Feb 15;37(1):30). The authors discussed the therapeutic value of T-DM1 in treating serous endometrial cancer. Actually, the same research group published another paper highlighting the potential of the novel anti-cancer compound SYD985 in serous endometrial cancers expressing moderate/low or heterogeneous HER2 expression, and SYD985 is 10- to 70-fold more potent than T-DM1 (Mol Cancer Ther. 2016 Aug;15(8):1900-9). Thus, this report should be shown to outline some future directions for the development of new treatment strategies against serous endometrial cancers. One way to overcome resistance to single-agent targeted therapy may be the use of drug combinations. Of note, the combination of the two inhibitors (Neratinib and Taselisib) caused a stronger and long-lasting growth inhibition in both serous endometrial cancer xenografts when compared with single-agent therapy (Mol Cancer Ther. 2015 Nov;14(11):2519-26). This dual HER2/PIK3CA blockade may represent a novel therapeutic option for serous endometrial cancer patients, and was not included in this review. Considering the aggressive nature of serous endometrial cancer cells, identifying the early molecular events in serous endometrial cancer carcinogenesis would be critical. For example, HMGA2 is expressed in serous endometrial intraepithelial carcinoma (Histopathology. 2012 Mar;60(4):547-53; Am J Cancer Res. 2016 Jan 15;6(2):249-59). In addition, GLUT1 is expressed in the vast majority of serous endometrial cancer and serous endometrial intraepithelial carcinoma, suggesting a biological role of GLUT1 during the early steps of carcinogenesis in serous endometrial cancers (Histopathology. 2006 Jul;49(1):75-81; Int J Surg Pathol. 2017 Aug;25(5):389-396). The importance of GLUT1 and HMGA2 in the early stage of serous endometrial cancer should be shown in Figure 1.

Minor issues

It would be better to give the readers a whole picture (such as a Figure or Table) showing the top mutated genes by serous endometrial cancer tissues by analyzing the data from the COSMIC database or the cBioPortal database. Line 413: PD-L1 is known as CD274, but not CD279.

Author Response

Thank you so much for your comments. Please see the attachment. 

Reviewer 2 Report

I doubt that endometrial uterine cancer is the most gynecological malignancy! What’s lacking in the introduction is a through literature review to introduce this topic to the scientific and clinical readers; e.g. a background with global epidemiological data has to be written.

Under question in this investigation are type 1 and 2 endometrial cancers; the molecular signatures are reviewed and p53 immunostainings: both parts are not well investigated let alone described and analyzed. For instance, what were the p53 staining patterns like? Many pathways are touched upon but how do they communicate? This part is really interesting but mainly descriptive for each individual pathway not really providing the big picture. Likewise, other interesting topics are discussed but not further developed: e.g. regarding mitochondrial functions: “After systematically analyzing 271 EECs from TCGA by bioinformatics, Liu et al found genes involved in the TCA cycle were decreased compared with normal controls” is all we read about the topic; what bioinformatics tools were used what was the outcome? Clinical, molecular, survival data?

Table 1 should include references where results were obtained.

Figure 2 is only based on 3 references; this seems scarce.

Overall the paper reads well but mostly is a long laundry list of many known cancer pathways without pulling the strings together. Which are the highlights? This is the reason why I am little enthusiastic about the contents of the paper. More meta-analytical and critical approaches would have been appreciated. 

Author Response

Thanks for the comments and Please see the attachment. 

Reviewer 3 Report

I read with great interest the paper entitled Pathogenesis and clinical management of uterine serous carcinoma. I think that some issue should be better clarified.  

Lines 40-42, you wrote:

Type II tumors (nonendometrioid carcinomas), including serous carcinoma, clear-cell carcinoma, and grade 3 endometrioid EC (EEC), are characterized by poorly differentiated histology, and deep migration/invasion…”

Type II endometrial cancers comprises non-endometrioid subtypes such as serous, clear cell and undifferentiated carcinomas, as well as carcinosarcoma/ malignant-mixed Müllerian tumour  (Colombo N, et al, ESMO-ESGO-ESTRO Endometrial Consensus Conference Working Group. ESMO-ESGO-ESTRO Consensus Conference on Endometrial Cancer: diagnosis, treatment and follow-up. Ann Oncol. 2016;27:16-41).

Moreover, grade 3 endometrioid carcinomas are characterized by particular heterogeneity. In some cases, they can show unequivocally endometrioid morphologic features in other cases they are histologically ambiguous making the distinction between serous-like grade 3 endometrioid and serous carcinomas difficult or impossible. It well known that interobserver variability in classifying endometrial carcinomas remains problematic, particularly in the subset of high-grade carcinomas, including the grey zone between high-grade endometrioid and serous carcinomas.

So I think that endometrial cancer classification and difficulties in distinguish  the different types should be clearly reported.

(Soslow RA, at al. Endometrial Carcinoma Diagnosis: Use of FIGO  Grading and Genomic Subcategories in Clinical Practice: Recommendations of the International Society of Gynecological Pathologists. Int J Gynecol Pathol. 2019;38:S64-S74. Goebel EA, Vidal A, Matias-Guiu X, Blake Gilks C. The evolution of endometrial carcinoma classification through application of immunohistochemistry and molecular diagnostics: past, present and future. Virchows Arch. 2018 ;472:885-896.).

Lines 265, you wrote

“Currently, the most common approach for USC treatment is surgery followed by chemotherapy and radiotherapy [85]. Similar to ovarian cancer, the routine and comprehensive surgery and staging of USC includes total hysterectomy, bilateral salpingo-oophorectomy, bilateral pelvic lymphadenectomy, para-aortic lymph sampling, complete omentectomy, and peritoneal cytology [86]”.

According to international guideline para-aortic lymph sampling is not recommended because systematic para-aortic lymphadenectomy till to left renal vein is mandatory. Please revise.

Line 272, you wrote

“After surgery, adjuvant chemotherapy is routinely recommended to USC patients, with the purpose of killing the remaining cancer cells or preventing them from growing [90]”.

It should be reported that in case of serous endometrial cancer arise from a polyp and without myoinvasion, adjuvant therapy or only postoperative observation are equally recommended particularly in case of absent residual tumour in the uterus after complete surgical staging. In these patients adjuvant therapy seems not to improve survival so it might be avoided. 

(Mandato VD, et al. Uterine Papillary Serous Carcinoma Arising in a Polyp: A Multicenter Retrospective Analysis on 75 Patients. Am J Clin Oncol. 2019;42:472-480. Thomas MB, et al. Role of systematic lymphadenectomy and adjuvant therapy in stage I uterine papillary serous carcinoma. Gynecol Oncol. 2007;107:186-9).

Round 2

Reviewer 3 Report

Dear Authors,

Please revise Lines 241-243, you wrote

"Similar to ovarian cancer, the routine and comprehensive surgery and staging of USC includes total hysterectomy, bilateral salpingo-oophorectomy, bilateral pelvic lymphadenectomy, complete omentectomy, and peritoneal cytology [61].

You should add that systematic para-aortic lymphadenectomy till to left renal vein is mandatory in comprehensive surgical staging of USC   

Regards

Author Response

Thanks for your comment. We added "systematic para-aortic lymphadenectomy till to left renal vein" in line 239-241, Page 8 of the revised manuscript.